# Breeding seasonality of Tylopoda: Expected patterns in Old World camelids but an exceptional pattern in South American camelids

Nicole Pauli[1], Marco Roller[2], Marcus Clauss[1]*

**1** Clinic for Zoo Animals, Exotic Pets and Wildlife, Vetsuisse Faculty, University of Zurich, Zurich, Switzerland, **2** Karlsruhe Zoo, Karlsruhe, Germany

* mclauss@vetclinics.uzh.ch

## Abstract

Seasonal breeding is a common adaptation among mammals in seasonal environments, ensuring offspring is born under favourable conditions. As only conception can be controlled but not birth itself, a predictive cue for these favourable conditions is needed (e.g., photoperiodism). Seasonal breeding, however, can also be disadvantageous if there will be time spent waiting until the next breeding season starts. We hypothesized that Old World camelids (OWC; *C. bactrianus* and *C. dromedarius*) exhibit a less pronounced seasonal breeding than South American camelids (SAC; *L. glama, L. guanicoe, V. vicuna, V. pacos*), given the higher costs of strict seasonality in OWC due to their gestation length (OWC ~ 13 months, SAC ~ 11 months). Both groups are expected to be long-day breeders, so that births occur in spring. We assessed data on conceptions of Camelidae, both from literature and zoo populations, and included information on latitude of origin and wet season. All camelid species were able to breed year-round, with the domestic forms of SAC being less seasonally restricted. OWC showed tendencies to long-day breeding, whereas SAC unexpectedly showed tendencies to short-day breeding. Patterns of conception peaks of literature and zoo populations were similar, suggesting the (partial) influence of photoperiodism. Further, inter-individual differences, as described for some rodents, cannot be ruled out. The climatic conditions and vegetational growth pattern of the natural habitat of SAC might explain the surprising short-day breeding seasonality. It deviates from the global pattern insofar as the most beneficial environmental conditions occur in autumn.

## Introduction

Seasonal variations affect most mammalian habitats, and many species adapt to these fluctuations with seasonally alternating physiological functions, behaviours and morphologies [1,2]. These changes involve migration, coat, metabolism and growth,

**Data availability statement:** All relevant data are within the manuscript and its Supporting Information files.

**Funding:** The author(s) received no specific funding for this work.

**Competing interests:** The authors have declared that no competing interests exist.

but the adaptation observed the most is seasonal breeding activity [2]. In large mammals, successful reproductive function is mainly determined by long-term food availability, with late pregnancy and lactation being the most sensitive stages. Reproductive function in small mammals, by contrast, is more susceptible to acute unpredictable environmental changes, low temperature or acute food shortages [1].

The ultimate aim of seasonal breeding is to time birth in relation to favourable environmental conditions for the lactating mothers as well as for the newborn offspring. As birth itself and the in utero development can only be controlled in a very limited manner, it is the mating activity and conception that is subject to specific timing so that births happen under favourable conditions [1–3]. Mammals breeding seasonally are therefore dependent on predictive cues in advance of a favourable birth season. The best known predictive cue is photoperiodism [1]. The light-dark cycle is transduced into a circadian rhythm of melatonin secretion, which is part of a signalling pathway that leads to changes in reproductive hormones (review, e.g., Vivid and Bentley [4]). Using photoperiod as a predictor is advantageous in a periodically changing environment, but it would be disadvantageous if the climatic and nutrient factors change irregularly or are relatively constant [1]. If animals mate within the period of increasing daylength, they are categorized as long-day breeders; animals mating during the period of decreasing daylength are called short-day breeders [5].

Besides the advantage of better offspring survival, seasonal breeding may also be accompanied by additional costs: 'off-season' conceptions must be prevented so that offspring will only be born in the most favourable season [6]. The period between the birth of one litter and the conception of the next plus the gestation time was defined by Kiltie [6] as the "minimum possible interval between litters (MIBL)". This author explained that highly seasonal species with a MIBL of more than 12 but less than 24 months will lose reproductive opportunity if there are only few favourable months for conception, as there would be a delay into the next year until mating could occur again after a birth. The opposite applies to seasonal species with a MIBL equal to or just less than 12 months (or a whole multiple of one year): even under highly seasonal breeding, offspring can be produced basically each year or at a regular cycle without lost periods [6]. To stay within this limit, there is most likely a strong adaptive value to shorten gestation periods in highly seasonal species [3]. Species with a short MIBL can thus afford to be seasonal breeders without losing reproductive opportunity, whereas species with a MIBL greater than 12 months are expected to be less seasonal breeders, due to the cost of lost opportunity [6], unless the waiting time is systematically filled by lactation. Arguably, regardless of the theory presented by Kiltie [6], environmental conditions might nevertheless select for a seasonal physiology if neonates have a low chance of survival outside of the favourable season.

If we assume that birthing in spring is the most favourable time for offspring survival in any environment, we can predict the mating season of seasonally reproducing species if we know their gestation period (Fig 1). Mammals with an either very short gestation period or a gestation of about a year are supposed to breed in spring and early summer (long-day breeders), whereas mammals with a gestational period of about six months are expected to breed in autumn (short-day breeders) [1,2].

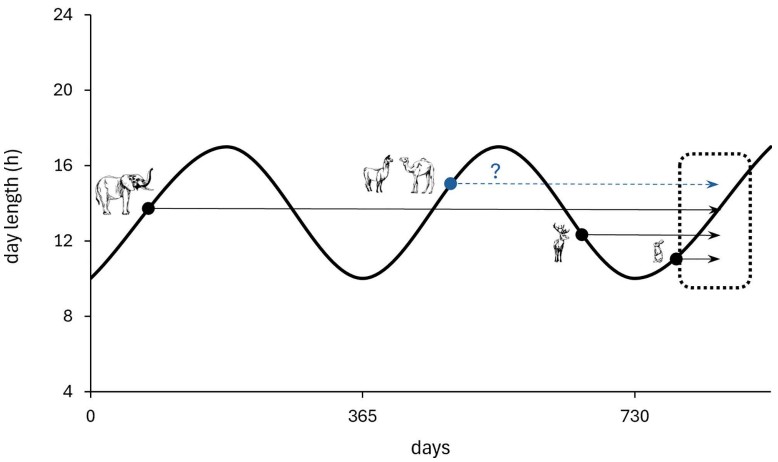

long-day vs. short-day breeders

**Fig 1. Schematic associations between gestation length, breeding season, photoperiod and assumed birthing season in spring.** Schematic presentation of the associations between gestation length (arrows), the onset of the breeding season (dots at the beginning of the arrows), a hypothei-cal photoperiod (day length) and an assumed favourable birthing season in spring (dashed rectangle). Seasonal breeders with a short gestation length (e.g., Snowshoe hares *Lepus americanus*) are long-day breeders, mating and giving birth within the same season [7]. Seasonal ruminants with gestation lengths of less than a year are short-day breeders with a mating season in autumn [8]. Elephants with gestation lengths of nearly two years tend towards long-day breeding [9]. Camelids with a gestation length of around a year are therefore suspected to be long-day breeders – if they had a seasonal repro-ductive pattern. Graph modified from Hufenus et al. [9]; animal icons drawn by Jeanne Peter.

Due to these factors, camelids are an interesting family regarding their assumed breeding seasonality, as they have gestation periods of just around a year: the South American camelids (SAC; *Lama glama, L. guanicoe, Viguna pacos, V. viguna*) less than 12 months; the Old World camelids (OWC; *Camelus dromedarius, C. bactrianus*) around 13 months (S1 Table in S1 File)). Both OWC and SAC are induced ovulators. Follicles develop in overlapping waves and the dominant follicles regress without ovulation. This means females can remain in a prolonged receptive state until copulation triggers ovulation (reviewed in, e.g., Vaughan and Tibary [10], Skidmore [11] and Musa et al. [12]).

The phylogeny of camelid species is not yet fully understood, and several hypotheses were proposed. The guanaco is most likely the ancestor of the llama [13,14], but the situation for alpaca is more complex. Although indications have been found that vicunas are the ancestors of alpaca [14], recent phylogenetic studies suggest that alpaca have derived in some part from both *Lama* and *Vicugna* clades and hybridisation played a role [13,15]. In OWC, the dromedary camel, the Bactrian camel and the wild *Camelus bactrianus ferus* are thought to be three separate species, or subspecies, with an estimated divergence dating back to the Pleistocene, which excludes the wild Bactrian camel as a direct ancestor of the domestic Bactrian camel or dromedary camel [16].

Assuming spring births are most favourable, we can hypothesize that camelids should be long-day breeders. This seasonality is thought to be an adaptive strategy that aligns calving with periods when environmental conditions are more favourable for calf survival, such as better forage quality and milder climate. This timing is especially important because, unlike many other ungulates, camelids do not lick their newborns dry; calving during milder seasons helps prevent hypo-thermia and supports neonatal thermoregulation when the calf is still wet after birth [10,17].

Further, we expect SAC to be more seasonal than OWC, as they have a MIBL of just less than or around 1 year. OWC in contrast, will have a MIBL of more than a year and are therefore expected to be less seasonal in their breeding; under strictly seasonal breeding, they would theoretically lose a full reproductive cycle every other year.

We reviewed the literature on the seasonality of reproduction in all six camelid species with a focus on free-ranging or semi-captive populations. Further, to evaluate a potential photoperiodic cueing, we also assessed data on breeding seasonality in captive camelid populations.

## Methods

We searched for relevant literature using Google Scholar, using keywords such as breeding seasonality, mating season, birth season or reproduction, both for the Camelidae family and each of the six species. Additionally, publications citing and cited in literature thus identified were reviewed. Information on the seasonal distribution of mating or birthing, latitude of origin, and start and end dates of wet season were gathered (Tables 1–6). Studies on photosensitivity (e.g., blindfolding or melatonin treatments) were also collected and are presented in the Supplementary Material.

If only birth dates were available, we calculated these dates back to conception dates, using 13 months for OWC and 11 months for SAC, respectively [18–23] (S1 Table in S1 File). Only the month of mating/birthing events was considered, as most sources did not report exact dates. For instance, birth on November 15th was assigned to a conception in December for SAC or to October for OWC. In some cases, data referred to the measurement of hormone levels [24–28], ovarian activity [23,25,29,30], testicular measurements [27,28,31] or a calculated conception date by means of embryo measurements [29,32]. If literature sources did not specify whether it described a mating or a birth season but just used the terms 'breeding season' or 'reproductive season' generically, we stated this in the same terms. When data was given as graphs and percentages instead of exact monthly case numbers, we extracted the underlying numerical data using WebPlotDigitizer (https://automeris.io/WebPlotDigitizer/).

The annual seasons were defined for the Northern Hemisphere as spring (March-May), summer (June-August), autumn (September-November), and winter (December-February). For the Southern Hemisphere, spring is September to November, and the subsequent seasons follow in the same 3-month-sections as in the Northern Hemisphere. Whenever a season refers to the Southern Hemisphere, it was labelled "austral".

The latitude of origin was either mentioned in the publication or obtained using Google Latitude Finder (http://www.latlong.net/) by searching for the described location of the study. In a few cases, we used country or regional averages or the mean latitude of several study locations when the results were only given as combined data [33–35]. In one case, the latitude of the main author's institution was used as no other latitudinal information was available [30]. In Schmidt [36], data was collected from 29 zoos on the Northern Hemisphere. For most camelids, the contributing zoos remain unknown, therefore no latitude was assigned.

Whenever available, data on wet season was taken from the same publication; when missing, we searched for studies on monthly precipitation for these locations.

For comparison with populations held in zoos, we analysed data on births of camelids provided by Species360 (www.species360.org). This international non-profit organization manages a database of wild animals held in zoos. Data from zoos on the Southern hemisphere was converted first by a six-month shift. Monthly births ratios were calculated as percentages of the total of all registered births, and converted to conceptions as explained above (Figs 2–7).

Upon researching the reproductive seasonality of camelids, several publications were found in which authors mentioned birth or mating seasons without specifying the corresponding data (no mentions of sample size or the method and duration of observation) or lacking observations for a full year (without giving a satisfactory explanation why this specific period of time was chosen or if there were breeding events during the remaining time of the year). Such publications were not included in Tables 1–6 but are mentioned in the supplement text (S1–S6 Text in S1 File).

In some cases, we decided to make exceptions to these criteria and included publications in Tables 1–6 although they were lacking a sample size, or observations did not cover a full year. In guanacos for example, in most studies an

exact sample size was not given and observations were based on populations or herds, without specifying the number of animals, matings or births [37–42]. If a reasonable explanation for a discontinuation of data collection of less than a year was given, we included this publication in our data collection (e.g., Deen et al. [31] described a clear onset and decline until cessation of the breeding activity, although the study only covered the months December to May). Further, studies collecting data through questionnaires filled out by camel owners were included [29,43–51]. These datasets often included a precise number of owned animals but did not give an exact quantity of births or matings. In some cases, the authors presented the data from questionnaires as a mixed dataset with additional data from direct observations or an experimental setup [29,44,47].

## Results

### Vicugna vicugna

Publications not summarized in Table 1 generally indicate mating and birth in autumn (short-day season) (S1 Text in S1 File). The detailed studies presented in Table 1 also show a tendency to mating season accumulation in the autumn months. In the Southern Hemisphere, studies from wild populations [52–55] as well as from research facilities [24,56] or semi-captive populations [57] reported mating mainly from March to May (austral autumn); some described a mating seasonality a bit earlier starting in December or January [24,55] and terminating in March or April [24,54–56]. The mating season of vicunas shows a tendency to start in the second half of, or with the end of the rainy season. Northern Hemisphere data [36,58] suggests a similar autumnal pattern; however this data is very sparse and with only low numbers of observations. Overall, vicunas are clear short-day breeders and mate around the autumn equinox.

Conceptions in vicuna populations from zoos (Fig 2) also accumulate around the autumn equinox, though a few conceptions were distributed throughout the rest of the year. Captive vicuna should be considered mainly short-day

**Table 1. Distribution of conceptions and conception peaks in vicunas** *(in relation to the rainy season, the long-day or short-day period).*

| Latitude | Month | | | | | | | | | | | | n | d | Source |
|---|---|---|---|---|---|---|---|---|---|---|---|---|---|---|---|
| | J | F | M | A | M | J | J | A | S | O | N | D | | | |
| | days getting longer | | | | | | days getting shorter | | | | | | | | |
| 51.32 | | | | | | x | x | | x | | | | 3 | 1 | °a |
| 50.49 | | x | | | | | | x | x | x | | x | 7 | 1 | °b |
| | days getting shorter | | | | | | days getting longer | | | | | | | | |
| −14.40 | | | x | xx | x | | | | | | | | 21 | 2 | ~c |
| −14.40 | | (x) | xx | xxx | xx | (x) | (x) | (x) | | | | | 190 | 1 | ~c |
| −16.02 | | | x | xx | x | | | | | | | | 36 | 1, 2 | ~d |
| −18.10 | | | x | x | | | | | | | | | 37 | 1 | ~e |
| −18.15 | x | x | x | | | | | | | | | | 8 | 2, 5 | -f |
| −18.28 | | (x) | x | xx | x | | | | | | | | – | 1 | +g |
| −21.50 | x | x | x | x | | | | | | | | x | 3 grps | 2 | ~h |
| −22.51 | | | xx | x | x | | | | | | | | 16 | 1 | -i |
| −22.51 | | x | x | | | | | | | | | | 7 | 2 | -i |

Latitude in [°]; x to xxx = number of conceptions increasing, xxx = conception peaks; (x) only a marginal number of data compared to the rest of the year; light grey squares = rainy season/wet season; *n* = number of observed births/matings/studied populations (- = number not indicated); *italic = most probably too little data.*

Sources: + semi-captive, ° zoo, ~ wild, – research facility, *d* = type of data: [1] births, [2] mating records, [5] hormone measurements.

a [58], b [36], c [52], d [53], e [54], f [24], g [57], h [55], i [56].

Sources rainy/wet season: from the corresponding publication on mating data if not stated otherwise; publication f [54].

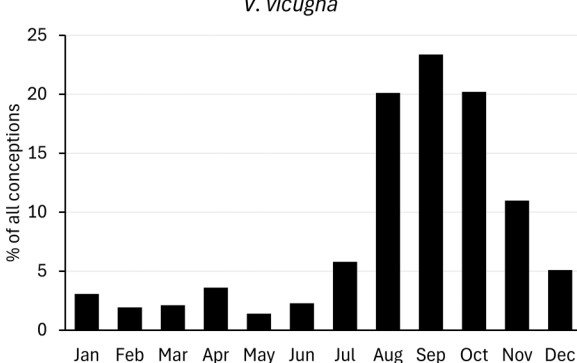

**Fig 2. Distribution of conceptions (1138 cases) in captive vicuna populations.** Long-day period: January to June. Data from populations on the Southern Hemisphere were included and corrected accordingly.

breeders with around 85% of conceptions registered in the short-day period. Thus, births accumulate in summer and early autumn.

## Vicugna pacos

Publications not included in Table 2 generally indicate mating and birth in autumn (short-day season) and some in summer. Further, a year-round mating is possible when female and male alpacas are housed separately and only brought together for mating purposes (S2 Text in S1 File).

Most studies in Table 2 of the Southern Hemisphere show a clear mating season for alpacas during the short-day period (austral autumn), primarily from December/ January to April/ May [18,19,59,60]. Two publications noted year-round mating [46,60]. Agramonte and Leyva [60] observed matings throughout the year with a concentration of births from December to April, whereas Hinojosa et al. [46] found two different mating seasonalities in alpaca herds in close proximity. The latter large-scale study interviewed 224 alpaca farmers owning more than 100 animals per herd;140 farmers reported continuous mating, and the males walked with the females all year round. The remaining 84 farmers observed seasonal mating from January to March (austral summer). Unfortunately, it remains unknown whether the males in these seasonal breeding herds ran with the females throughout the year or only during the summer months [46]. This observation is in contrast with the statement above of Fernández-Baca et al. [61] where a seasonal breeding was observed specifically in herds which kept females and males together all year round.

In the Northern Hemisphere, tendencies of birth peaks are uncertain, due to limited data (two publications with small numbers of observations).

Overall, alpacas presumably are short-day breeders, with most conceptions between the austral summer solstice and the austral autumn equinox. For the presented locations in Table 2, this aligns with the summer rainy season (December-March), with a mating peak in the second half of the rainy season.

When assessing the reproductive season of alpacas kept in zoos, the numbers of conceptions increase from May onwards, peak in July and August, and decline to a nadir in February (Fig 3). Conceptions in zoos clearly accumulate in the short-day period of the year (68% of conceptions), supporting the classification of alpaca as short-day breeders. Consequently, births of *V. pacos* in zoos are mainly observed in summer and autumn.

**Table 2. Distribution of conceptions and conception peaks in alpacas** *(in relation to the rainy season, the long-day or short-day period).*

| Latitude | Month | | | | | | | | | | | | n | d | Source |
|---|---|---|---|---|---|---|---|---|---|---|---|---|---|---|---|
| | J | F | M | A | M | J | J | A | S | O | N | D | | | |
| | days getting longer | | | | | | days getting shorter | | | | | | | | |
| 51.32 | | | x | x | x | | xx | | x | | | | 9 | 1 | °a |
| ? | x | (x) | | (x) | (x) | (x) | (x) | (x) | x | xx | x | x | 27 | 1 | °b |
| | days getting shorter | | | | | | days getting longer | | | | | | | | |
| −13.09 | x | x | x | | | | | | | | | | 84 | 3 | *c |
| −13.09 | x | x | x | x | x | x | x | x | x | x | x | x | 140 | 3 | *c |
| −14.28 | x | xx | xx | xx | x | (x) | (x) | (x) | (x) | (x) | (x) | (x) | 105 | 1, 2 | *d |
| −14.30 | x | x | x | x | x | | | | | | x | x | 306 | 2 | -e |
| −14.59 | xx | xxx | xxx | x | (x) | (x) | | | (x) | (x) | | x | 6533 | 1 | -f |
| −15.47 | xx | xx | xx | x | x | | | | | | | x | 707 | 2 | -g |
| −15.47 | x | xx | xx | x | x | | | | | | | x | 617 | 1 | -g |

Latitude in [°]; x to xxx = number of conceptions increasing, xxx = conception peaks; (x) only a marginal number of data compared to the rest of the year; light grey squares = rainy season/wet season; n = number of observed births/matings/studied populations; *italic = most probably too little data.*

Sources: *(milk) farm/ pastoral herd/unspecified ranch, °zoo, – research facility.

d = type of data: [1] births, [2] mating records, [3] birth or mating records obtained through questionnaires.

a [58], b [36], c [46], d [60], e [18], f [59], g [19].

Sources rainy/wet season: publication c [62], d [63], e [64], f [65], g [19].

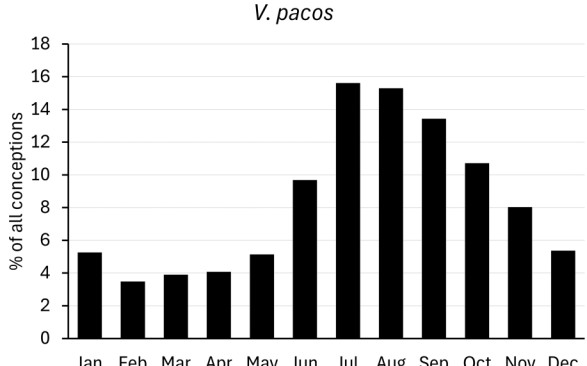

**Fig 3. Distribution of conceptions (3387 cases) in captive alpaca populations.** Long-day period: January to June. Data from populations on the Southern Hemisphere were included and corrected accordingly.

## Lama guanicoe

Publications not included in Table 3 indicate various seasonal mating seasons throughout the year with no common pattern (S3 Text in S1 File). In contrast, for the Southern Hemisphere, the studies presented in Table 3 indicate a short mating season (2–3 months) during summer. An exception was observed by Ortega and Franklin [42] where the mating season lasted for six months.

Eight of the ten studies from the Southern Hemisphere were conducted between 51°S and 53.5°S, either in Torres del Paine National Park in Chile or on the island of Tierra del Fuego of Argentina and Chile. In Torres del Paine National Park, the mating season started at the end of austral spring/ beginning of austral summer in November or December [39,40,66,67] with the exception of one study where mating started in October [42]. The mating season is followed by the rainy season

**Table 3. Distribution of conceptions and conception peaks in guanacos** *(in relation to the rainy season, the long-day or short-day period).*

| Latitude | Month | | | | | | | | | | | | n | d | Source |
|---|---|---|---|---|---|---|---|---|---|---|---|---|---|---|---|
| | J | F | M | A | M | J | J | A | S | O | N | D | | | |
| | days getting longer | | | | | | days getting shorter | | | | | | | | |
| 51.40 | | x | x | x | x | x | x | xx | x | x | x | | 141 | 1 | °a |
| 51.32 | | | | | | x | x | | | x | | | 3 | 1 | °b |
| ? | x | x | (x) | (x) | (x) | xx | xxx | xxx | xx | xx | x | (x) | 270 | 1 | °c |
| | days getting shorter | | | | | | days getting longer | | | | | | | | |
| −34.54 | x | x | (x) | (x) | | (x) | (x) | | x | (x) | | | 16 | 1 | °d |
| −39.43 | x | x | | | | | | | | | | x | 5 | 1 | -e |
| −51.00 | x | | | | | | | | | | x | xx | herds | 1, 2 | ~f |
| −51.03 | x | | | | | | | | | | | xx | 170 | 2 | ~g |
| −51.03 | x | x | | | | | | | | | | xx | herds | 1, 2 | ~h |
| −51.03 | (x) | (x) | | | | | | | | | | xx | 44 | 2 | ~i |
| −51.03 | x | x | x | | | | | | | x | x | x | 1 pop | 1, 2 | ~k |
| −53.04 | x | x | x | | | | | | | | | | herds | 1 | ~l |
| −53.40 | x | x | x | | | | | | | | | | 8 grps | 1 | ~m |
| −53.50 | x | xx | x | | | | | | | | | | herds | 1, 2 | ~n |

Latitude in [°]; x to xxx = number of conceptions increasing, xxx = conception peaks; (x) only a marginal number of data compared to the rest of the year; light grey squares = rainy season/wet season; *n* = number of observed births/matings/studied populations or herds; *italic* = most probably too little data.

Sources: ° zoo, ~ wild, – research facility.

*d* = type of data: [1] births, [2] mating records.

a [35], b [58], c [36], d (Lancaster in litt., cited in [36]), e [72], f [40], g [66], h [39], i [67], k [42], l [38], m [41], n [37], sources rainy/wet season: publications e [73], f-k [68], l-n [69].

[68]. Further South on the island of Tierra del Fuego, the mating seasonality of guanacos started later – in January (austral summer) [37,38,41]. Here, no clear rainy season can be observed as precipitation spreads evenly over the year [69].

Publications with data sourced from zoos show a continuous distribution of mating throughout the year [35,36,58], with a mating peak in summer.

In conclusion, these data cannot clearly decide whether free-roaming guanacos are long-day or short-day breeders; however, most mating is observed after the austral summer solstice in the short-day period. Guanacos tend to breed continuously to a certain extent when housed in zoos.

Some cues even indicate a direct link of the mating seasonality of guanacos to the austral summer solstice itself. Jurgensen [67] observed an increase in copulations after December 21st in Chile: of the total 44 copulations observed, 34 of them happened in December and 60% of those between December 22nd and December 28th [67]. Similarly, Bank et al. [66] reported peak mating in the last two weeks of December in Chile. Unfortunately, this data was grouped in half-month intervals, limiting precise interpretation regarding the solstice on December 21st. Other observations in Chile noted the start of the birth season after November 19th [39], which – given a 11.5-month gestation (S1 Table in S1 File) – suggests mating around the solstice. However, including the duration of ovarian waves (15.1±4.2 days [70]) and follicular growth (7.0±2.4 days [70]) under the assumptions that the ovaries of guanacos are less active in the months outside the mating season [71] and that they would rely on the southern summer solstice for activating breeding activities and starting a new follicular wave, the start of the mating season of these mentioned publications lies rather too close to the solstice event. Thus, the assumption that a longer long-day period could act as a trigger is also plausible.

When assessing the conception of guanacos held in zoos (Northern Hemisphere), a rise in conceptions can be observed from May to July, with most conceptions taking place in July (summer) (Fig 4). Compared to the data in Table 3,

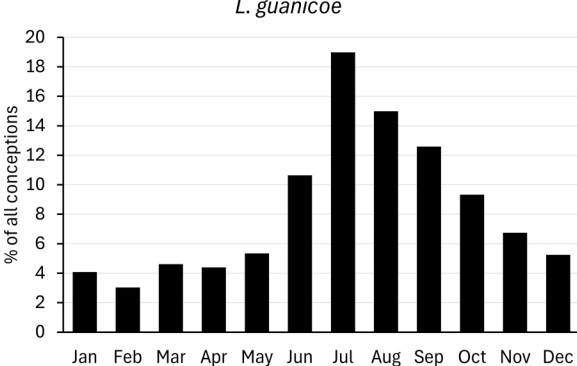

**Fig 4. Distribution of conceptions (2208 cases) in captive guanaco populations.** Long-day period: January to June. Data from populations on the Southern Hemisphere were included and corrected accordingly.

conceptions of captive guanacos occur year-round, but a peak after the summer solstice can still be seen, although broader and stretching into autumn (September to November). With 67% of the conceptions in the second half of the year, captive guanacos can be considered short-day breeders capable of year-round reproduction, with births peaking from late spring to early autumn (May-September).

### Lama glama

Publications that are not summarized in Table 4 generally indicate mating and birth in autumn (short-day season), though some reported better sperm quality in winter (S4 Text in S1 File).

Only a few detailed studies were found regarding the breeding seasonality of llamas, and half of them were birth records of zoos (Table 4). These publications show a distribution of parturitions throughout the whole year for both hemispheres [35,36,58,74]. For the other publications, an accumulation of mating was found during the summer and autumn months (December/January to April in the Southern Hemisphere [44,45,75]; June to September in the Northern Hemisphere [76]). To a certain extent, the same accumulation of conceptions during these summer and early autumn months can be found in two publications with data obtained from zoos [35,36].

In the study with the biggest data set of more than 2400 birth records from the Quimsachata Experimental Station in Peru [75], nearly half of all records were in February (1078 births; austral summer). Therefore, most mating had happened in March (austral autumn). For January, March and April, recorded birth numbers were 644, 515 and 141 respectively. For the remaining months only 39 birth records were collected in the whole data set covering 10 years.

In conclusion, llamas show a tendency to short-day breeding, with a less pronounced peak than guanacos. The mating season in llamas is slightly longer than in guanacos, and peak seasonality, if present, was between the summer solstice and the autumn equinox, whereas in guanacos peak mating season lies closer to the summer solstice.

We further need to note that all non-zoo data from the Southern Hemisphere come from latitudes between the equator and the Southern Tropic. Where data on rainy season was included, the peak mating season seems to coincide with the second half of the rainy season.

Data from zoo llama populations (Fig 5) shows a distribution of conceptions throughout the whole year with a minor peak in July and an accumulation between June to September. This is in agreement with data from zoo populations in Table 4 [35,36]. In conclusion, captive llama populations breed year-round but show a minor accumulation during the short-day season (56% of conceptions) with births most pronounced from late spring to summer (May to August).

**Table 4. Distribution of conceptions and conception peaks in llamas** *(in relation to the rainy season, the long-day or short-day period).*

| Latitude | Month | | | | | | | | | | | | *n* | *d* | Source |
|---|---|---|---|---|---|---|---|---|---|---|---|---|---|---|---|
| | J | F | M | A | M | J | J | A | S | O | N | D | | | |
| | days getting longer | | | | | | days getting shorter | | | | | | | | |
| 51.40 | | x | x | x | x | x | xx | xx | xx | x | x | | 78 | 1 | °a |
| 51.32 | x | x | x | xx | x | x | x | x | xx | xx | x | x | *38* | 1 | °b |
| 51.32 | | | | | x | | | x | x | | | | *11* | 1 | -c |
| ? | x | x | x | xx | xx | xx | xxx | xxx | xxx | xx | xx | x | 507 | 1 | °d |
| | days getting shorter | | | | | | days getting longer | | | | | | | | |
| −15.45 | (x) | xx | xxx | xx | x | (x) | (x) | (x) | (x) | (x) | (x) | (x) | 2417 | 1 | -e |
| −17.40 | x | | | | | | | | | | | x | herds | 3 | *f |
| −21.24 | xx | xx | xx | x | (x) | | | | | | (x) | x | herds | 3 | *g |
| −25.44 | x | x | x | x | x | x | x | x | | x | x | x | *40* | 1 | °h |

Latitude in [°]; x to xxx = number of conceptions increasing, xxx = conception peaks; (x) only a marginal number of data compared to the rest of the year; light grey squares = rainy season/wet season; *n* = number of observed births/matings/studied populations; *italic* = most probably too little data.

Sources: *(milk) farm/ pastoral herd/unspecified ranch, °zoo, – research facility, *d* = type of data: [1] births, [3] birth or mating records obtained through questionnaires.

a [35], b [58], c [76], d [36], e [75], f [44], g [45], h [74].

Sources rainy/wet season: from the corresponding publication on mating data if not stated otherwise; publication c: no reliable data on rainy season could be found.

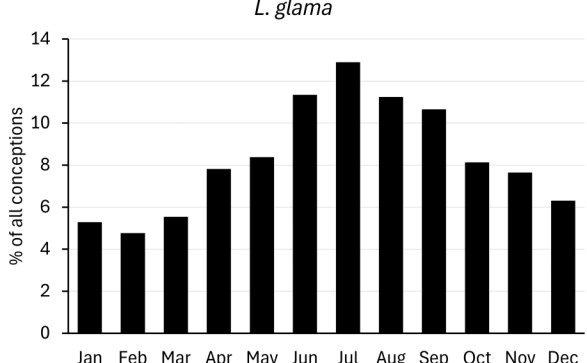

**Fig 5. Distribution of conceptions (4786 cases) in captive llama populations.** Long-day period: January to June. Data from populations on the Southern Hemisphere were included and corrected accordingly.

## Camelus bactrianus

Publications not listed in Table 5 indicate a mating season mainly in winter and early spring (long-day season), though some report mating already in late autumn (S5 Text in S1 File). The studies in Table 5 were all carried out with captive, domesticated Bactrian camels. No publication could be found providing sufficient data on truly wild Bactrian camels (*Camelus bactrianus ferus*). Four of the eight studies presented are based on zoo observations [34–36,58].

Unlike the SAC, which show a mating season in (austral and boreal) summer and autumn, the mating season of Bactrian camel is in winter and early spring. However, two publications report a mating season starting in later autumn [47,77]. All available data are from latitudes north of the Tropic of Cancer (23°26' N). Within these latitudes, a slight tendency to an earlier start and earlier ending of the mating season was observed the closer the study population is based to the equator.

**Table 5. Distribution of conceptions and conception peaks in Bactrian camels** *(in relation to the rainy season, the long-day or short-day period).*

| Latitude | Month | | | | | | | | | | | | *n* | *d* | Source |
|---|---|---|---|---|---|---|---|---|---|---|---|---|---|---|---|
| | J | F | M | A | M | J | J | A | S | O | N | D | | | |
| | days getting longer | | | | | | days getting shorter | | | | | | | | |
| 51.40 | x | xx | x | x | x | | | | x | | | | | 89 | 1 | °a |
| 51.32 | | x | x | | | | | | | | | | | *6* | 1 | °b |
| 49.44 | | xx | x | x | | x | | | | | | | x | *12* | 1 | °c |
| 37.21 | x | x | x | x | | | | | | | (x) | x | | 130 | 2 | *d |
| 37.21 | x | x | x | x | | | | | | | | (x) | | 69 | 2, 6 | *d |
| 34.10 | x | x | x | | | | | | | x | x | x | | herds | 3 | -e |
| 33.30 | x | x | x | | | | | | | x | x | x | | 54 | 2 | *f |
| ? | x | xx | x | x | (x) | (x) | (x) | | | | | (x) | | 166 | 1 | °g |

Latitude in [°]; x to xxx = number of conceptions increasing, xxx = conception peaks; (x) only a marginal number of data compared to the rest of the year; light grey squares = rainy season/wet season; *n* = number of observed births/matings/studied populations; *italic* = most probably too little data.

Sources: *(milk) farm/ pastoral herd/unspecified ranch, °zoo, – research facility.

*d* = type of data: [1] births, [2] mating records, [3] birth or mating records obtained through questionnaires, [6] ovarian activity.

a [35], b [58], c [34], d [23], e [47], f [77], g [36].

Sources rainy/wet season: publication d [79], e + f [80].

No data is available for the Southern Hemisphere. For all study populations, the mating season starts around a month after the end of the rainy season, which lies in summer and autumn.

In conclusion, Bactrian camels are long-day breeders. Some cues were found that the winter solstice could influence the breeding activities. The onset of the ovarian follicle maturation phase was between December 24th and January 9th in a study examining 5 female Bactrian camels through ultrasonography. The average length of the maturation phase was 10 days [78]. Similarly, Chen and Yuen [23] observed 87% of the female Bactrian camels entering the breeding season in the first half of January upon rectal palpation.

Data on conceptions of zoo *C. bactrianus* populations show a sharp peak in February (winter), accounting for one third of all conceptions (Fig 6). Numbers of conceptions decline towards June/ July, with only 10% of conceptions during the second half (short-day period) of the year. In conclusion, captive Bactrian camels are long-day breeders with most births occurring in spring (March).

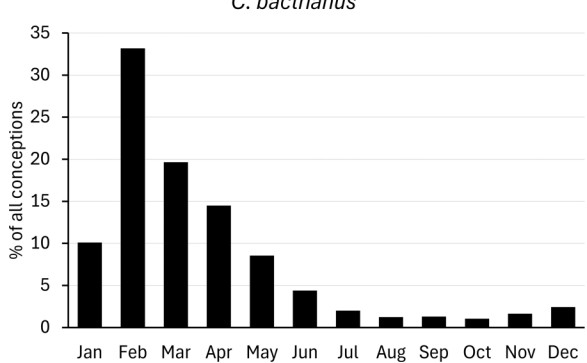

**Fig 6. Distribution of conceptions (3334 cases) in captive Bactrian camel populations.** Long-day period: January to June. Data from populations on the Southern Hemisphere were included and corrected accordingly.

### Camelus dromedarius

Publications not summarized in Table 6 generally indicate a breeding season from late autumn/ early winter (November/ December) to winter/spring (February to May), often without specifying whether birthing or mating was observed. In a few publications, the breeding season extends into the summer months, and peak mating season was in winter, if observed. In conclusion, these breeding seasons fall in the short-day season, though sometimes start at the end of the long-day season (S6 Text in S1 File).

Various studies with sufficient data (Table 6) were available in dromedary camels, most of them from latitudes on the Northern Hemisphere. North of the Tropic of Cancer, a seasonal mating from November/December to April/May was common, whereas south of it a tendency to year-round mating was observed. However, five studies north of the Tropic of Cancer (latitudes between 25° N and 33° N) indicate a year-round or nearly year-round mating [32,81–84]. Benaissa et al. [84] and Elwishy and Ghoneim [32] obtained their data from measuring foetuses in slaughtered camels. Benaissa et al. [84] found an accumulation of conceptions during winter (October to March) in Algeria, with a peak in January and smaller numbers of conceptions throughout the remaining year. In contrast, Elwishy and Ghoneim [32] found a summer mating season (May to August) in Egypt, with a peak in July and lower numbers of calculated conceptions scattered throughout the year.

This latter result stands alone in its explicitness, but we see a few similar results of concentrated breeding during the summer months by other authors. A second study from Egypt observed seasonal mating in winter (December to March) and a peak in January [83]. The same authors also reported birth data, where a peak in July was observed (besides a winter birth season) [83]. With a gestation length of 13 months (S1 Table in S1 File), this birth peak in July corresponds to a mating peak in June. Another example of intensified mating during the summer months is provided by Abbas et al. [49]. Based on questionnaires of camel pastoralists in Sudan, 70% of all matings were observed between April to July [49]. In Kenya, year-round mating with minor accumulations of intensified mating during spring and early summer (April to June) and late autumn and winter (October to December) was observed [85].

A somewhat unique breeding seasonality was described by Traoré [43] for Mali, as there is neither a peak in nor restriction of breeding activities during winter nor summer, nor is it a year-round breeding pattern. Instead, mating was observed in the short-day season from June to December (based on questionnaires).

For the Southern Hemisphere, only three studies were available. Two of them [51,85] gave a year-round breeding with no clear conception peak. The third study provides a small data set with only nine birth records [74].

The onset of the rainy season varies with latitude. In some regions, the beginning of the rainy and mating seasons coincide; at other latitudes the rainy season precedes the mating season by a few months. Another scenario is seen for the studies from India, where a summer monsoon rainy season is noted but the mating season is restricted to the winter months [27,31,77,86]. Close to the equator, even two rainy seasons per year were recorded and a tendency to intensified mating during these times can be observed [51,85].

Overall, dromedary camels cannot be classified as short-day or long-day breeders but apparently breed around both sides of the winter solstice. North of the Tropic of Cancer, they tend to mate seasonally during winter, though in a few cases a year-round mating was noted. South of the Tropic of Cancer, a year-round breeding can be observed.

In dromedary populations kept in zoos (Fig 7), 72% of conceptions were registered during the first half of the year (long-day period); peaking in February, declining towards July and increasing again in November/ December. Similarly, the literature data (Table 6) showed increased mating from November/December to March/April, although in some populations a peak (or second peak) was observed in June/July, and in populations close to the equator a continuous breeding was noted. In zoo dromedary populations (Fig 7), it remains unknown if these two latter observations can also be seen in (some) zoos. Overall, zoo dromedary populations mainly mate during the long-day period but are capable of a year-round breeding. Thus, most births were registered in late winter and spring (February to April/May).

**Table 6. Distribution of conceptions and conception peaks in dromedary camels** *(in relation to the rainy season, the long-day or short-day period).*

| Latitude | Month | | | | | | | | | | | | n | d | Source |
|---|---|---|---|---|---|---|---|---|---|---|---|---|---|---|---|
| | J | F | M | A | M | J | J | A | S | O | N | D | | | |
| | days getting longer | | | | | | days getting shorter | | | | | | | | |
| 51.32 | x | | x | | | | x | | | x | | | *4* | 1 | °a |
| 33.21 | xx | x | x | | | | | | | | x | xx | 60 herds | 3 | %b |
| 33.14 | xxx | xx | xx | x | x | x | x | x | x | xx | xx | xx | 198 | 4 | #c |
| 31.18 | xx | xx | xx | x | x | (x) | (x) | (x) | (x) | (x) | x | xx | 1 herd | 2 | -d |
| 31.18 | xx | xx | xx | (x) | (x) | xxx | (x) | (x) | (x) | (x) | x | xx | 1 herd | 1 | -d |
| 30.03 | x | x | x | x | xx | xx | xxx | xx | x | x | x | x | 579 | 4 | #e |
| 30.03 | x | x | x | x | | | | | | | x | x | 320 | 5, 6 | #f |
| 29.30 | x | x | x | x | | | | | | | | | *6* | 5, 8 | *g |
| 27.58 | xxx | xx | x | x | | | | | x | x | xx | xxx | 988 | 1 | -h |
| 27.58 | xx | xx | x | | | | | | x | x | x | | *4* | 5, 8 | -i |
| 27.58 | xx | xx | xx | xx | x | | | | | | x | | *4* | 2, 8 | -i |
| 27.58 | x | xx | xx | xx | xx | | | | | | x | | *4* | 7, 8 | -i |
| 27.58 | x | x | x | | | | | | | | x | | 71 | 2 | -k |
| 27.58 | x | xx | xx | xx | x | | | | | | x | | 203 | 2, 7, 8 | -l |
| 27.07 | xxx | xx | x | x | (x) | | (x) | (x) | (x) | x | xx | xxx | 685 | 1 | *m |
| 27.07 | xxx | xx | xx | x | (x) | | (x) | (x) | (x) | x | xx | xxx | 229 | 1 | *m |
| 25.41 | x | x | x | (x) | | | | x | xxx | xxx | xx | | *29* | 1, 2 | *n |
| 25.41 | xx | xx | x | x | | | | (x) | xx | xxx | xxx | | *17* | 1, 2 | *n |
| 25.0 | xx | xx | xx | xx | x | (x) | | (x) | x | x | xx | xxx | 4248 | 1 | *o |
| 23.35 | xx | x | x | (x) | | | | (x) | x | x | xx | | *8* | 5, 7, 8 | -p |
| 23.15 | xx | x | x | x | x | x | x | x | x | x | xx | xx | – | 3, 4, 6 | %#q |
| 16.00 | | | | | x | x | x | x | x | x | x | x | 100 herds | 3 | %r |
| 15.35 | | x | x | x | x | x | x | x | | | | | *5* | 6 | *s |
| 14.54 | x | x | (x) | xx | xx | xx | xx | (x) | (x) | (x) | (x) | x | 822 | 1, 3 | *t |
| 11.45 | xx | x | x | x | x | x | (x) | (x) | (x) | xx | xx | xx | 345 | 3 | *u |
| 1.44 | x | x | x | xx | x | x | x | | x | x | xxx | xxx | 88 | 1, 3 | %v |
| 0.28 | (x) | (x) | (x) | x | x | x | x | x | x | x | (x) | x | 115 | 1 | *w |
| 0.25 | x | x | xx | xx | xx | xx | x | x | x | x | xx | x | 179 | 1 | *w |
| 0.24 | | x | x | x | x | x | x | x | (x) | x | x | (x) | 62 | 1, 3 | *v |
| 0.03 | xx | x | x | x | x | x | x | x | x | x | x | x | 119 | 1 | *w |
| – | x | x | x | x | (x) | (x) | (x) | (x) | (x) | | (x) | (x) | *47* | 1 | °x |
| | days getting shorter | | | | | | days getting longer | | | | | | | | |
| −2.27 | x | x | x | xx | x | x | x | x | x | x | x | x | 102 | 1, 3 | *v |
| −3.00 | x | x | x | xx | xx | x | x | xx | x | x | x | xx | 223 | 1 | *w |
| −25.44 | | x | | x | | | x | x | | x | | | *9* | 1 | °y |

Latitude in [°]; x to xxx = number of conceptions increasing, xxx = conception peaks; (x) only a marginal number of data compared to the rest of the year; light grey squares = rainy season/wet season; n = number of observed births/matings/studied populations (- = number not indicated); *italic = most probably too little data.*

Sources: *(milk) farm/ pastoral herd/unspecified ranch, % nomadic, #slaughterhouse, °zoo, -research facility.

*d* = type of data: [1] births, [2] mating records, [3] birth or mating records obtained through questionnaires, [4] embryo measurements, [5] hormone measurements, [6] ovarian activity or [7] semen quality/structure of testes, [8] data only from male sources.

a [58], b [48], c [84], d [83], e [32], f [25], g [26], h [86], i [27], k [77], l [31], m [81], n [87], o [82], p [28], q [29], r [43], s [30], t [49], u [50], v [51], w [85], x [36], y [74].

Sources rainy/wet season: from the corresponding publication on mating data if not stated otherwise; publication d [88], e+f [89], g [90], h-l [91], m [92], n+o [93], q [94], s [95], t [96], v [97].

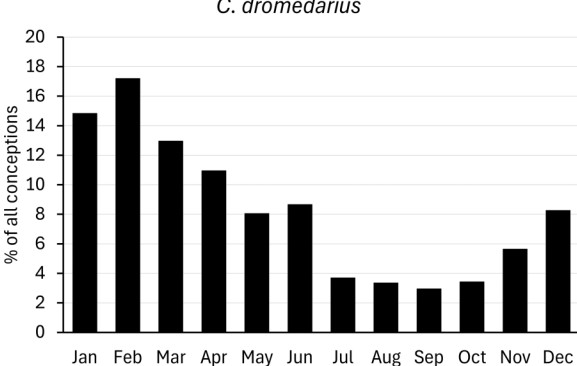

**Fig 7. Distribution of conceptions (1489 cases) in captive dromedary populations.** Long-day period: January to June. Data from populations on the Southern Hemisphere were included and corrected accordingly.

## Studies on photosensitivity

Several publications described increased reproductive activity around a month after administration of a melatonin implant in the off-breeding season, both in male and female dromedaries. Similar results were obtained when dromedaries were housed in artificial short-day photoperiod or when blindfolded, whereas reproductive activity decreased when simulating long days. These results would normally be interpreted as indications for a short-day breeding but could also be interpreted as preparation for reproductive activity towards the end of the short-day period. Publications on photosensitivity in Bactrian camels and SAC were sparse (S7 Text in S1 File).

## Discussion

We reviewed literature on reproductive seasonality of Camelidae and compared it to data of conceptions in camelids kept in zoos to evaluate potential photoperiodic cueing. Camelids have a gestation length of around a year (approximately 13 months in OWC and 11 months in SAC; S1 Table in S1 File). Based on considerations of Kiltie [6], we hypothesized that all camelids are long-day breeders as we assumed that spring is the most favourable season for offspring survival. OWC were expected to be less seasonal than SAC. Other authors suggested various breeding seasonality types for all camelids: long-day breeders (e.g., Chen et al. [98] Correa et al [99]), short-day breeders (e.g., Ainani et al. [100], Swelum et al. [101], Tibary and El Allali [102]), influenced by rainfall [100,102,103] or non-seasonal breeders [103,104].

In OWC, *C. bactrianus* showed a mating period from winter to spring, with a conception peak in February, both in the reviewed literature and in zoo populations. In zoo populations, minor numbers of conceptions were also registered during the short-day season. Dromedary camels could not be clearly classified as short- or long-day breeders; however, a tendency to long-day breeding was noted. This was more pronounced in zoo populations, in which most conceptions occurred in late winter and early spring. Both literature and zoo data revealed a capacity for year-round breeding in *C. dromedarius*, which appears to be more prevalent in the wild populations living closer to the equator. This tendency to long-day breeding was not directly supported by studies on effects of melatonin on breeding activities. Blindfolding and melatonin implants during the short-day season stimulate breeding activities in dromedary camels, which could be interpreted as an artificial acceleration of the short-day period, triggering a response occurring towards the winter solstice under natural conditions.

Unexpectedly, none of the SAC showed a clear long-day breeding season, neither in the reviewed literature nor in the data from zoo populations. Instead, whenever a distinct breeding season could be ascertained, it was during the short-day period. This was more pronounced in the non-domesticated species *V. vicugna* and *L. guanicoe* than in the domesticated

*V. pacos* and *L. glama*. As in OWC, zoo populations of SAC were capable of year-round breeding. Studies on melatonin and impact on reproductive physiology were nearly absent in SAC.

Thus, there are indications of a photoperiod-triggered breeding season in camelids. However, all camelid species were also capable of a year-round breeding. So far, no interpretation fully explains this mixed breeding pattern, even though a similar 'mixed' state of physiology has been suggested for other species. In elephants it was interpreted as an underlying photoperiod-triggered system influenced by body condition and stress or as an evolutionary transition stage [9]. In Siberian hamsters, two breeding phenotypes exist – one that is photoperiod-sensitive and will undergo gonadal regression when exposed to short days, and one that is not and will continue breeding [105]. Different responses to photoperiod changes have also been observed in deer mice and white-footed mice, both within a population as well as between populations of different latitudinal origins. However, in comparison to Siberian hamsters, the nonresponsive individuals in these mice still produce typical short-day melatonin patterns but do not exhibit the expected reproductive changes [105]. For camelids, the available data does not allow judging whether all animals are able to reproduce year-round but have a propensity for a certain season, or whether populations include distinct phenotypes. In the latter case, the (aseasonal) phenotype non-responsive to photoperiod might be inadvertently promoted under human care, and hence during domestication. However, as both scenarios would present similarly under human care, the difference between wild and domestic camelids does not indicate which scenario is more likely. Further studies following individual animals over long time periods are required to decide this. In addition, the induced ovulation in camelids leads to a prolonged receptive state of the female. Further studies should explore how this influences the mixed breeding patterns.

We argued that dromedary camels show a tendency to long-day breeding based on literature and copulations in zoo populations. Yet, blindfolding and melatonin implants (which could be understood as a simulation of a short-day period) stimulated reproductive functions, too. In sheep (short-day breeder) and hamsters (long-day breeder) a short-day response typical for the respective species was obtained by long-duration melatonin infusions (reviewed in [106]), which could lead us to the assumption that dromedary camels are in fact short-day breeders based on the response to melatonin implants. Alternatively, follicle maturation could be triggered by extremely short days so that mating can occur shortly after the winter solstice. More detailed studies into the photoperiodic response would be necessary for clarification.

As Kiltie [6] noted, species with a gestation of 11–12 months (e.g., horses and vicunas) are particularly interesting regarding the breeding seasonality, as in environments with short favourable seasons for offspring survival, the time cost of reproductive aseasonality increases greatly. These species thus benefit from a strictly seasonal breeding [6].

In contrast, OWC, with gestations over a year, would benefit from a less seasonal reproduction – in strictly seasonal breeding they would lose reproductive opportunity every other year. Strategies to reduce this "risk" could be (i) a varying length of gestation as observed by Nagy and Juhász [82]: gestation was nearly 20 days shorter in dromedary camels conceiving at the end of the breeding season compared to the beginning and middle of the breeding season; or (ii) a shift of reproductive focus towards (continuous) lactation to "fill" the time between two consecutive offsprings.

Our hypothesis that OWC would be less seasonal than SAC was not supported. Vicuna and guanaco were more seasonal than dromedaries and Bactrian camels. However, alpacas and lamas did not show a stricter seasonality than OWC. Further, Bactrian camels were more seasonal than dromedaries, despite sharing a gestation length of around 13 months.

S2 Table in S1 File gives an overview of the duration of lactation in camelid species. At a gestation period of 13 months, strictly seasonal OWC would not lose reproductive opportunity at lactation periods of 23 months. In dromedaries, lactation ranges 7–24 months, with the longest durations in non-pregnant females. A few studies described the time of cessation of lactation in pregnant dromedary camels in relation to the time of conception. If conception occurred within the first month postpartum, lactation persisted for around 10 months [107]. In contrast, if conception was later than within one month postpartum, lactation ceased within the next 2–4 months [108–110]. In conclusion, lactation length in dromedaries varies depending on herd management and time of conception after the previous birth. Therefore, regarding the potential

reproductive time lost between two seasons, dromedaries can be argued to benefit from a less seasonal breeding as they will not be able to "fill" the time between two parturitions with lactation in any case.

The limited data for Bactrian camels suggests a lactation of 10–16 months (S2 Table in S1 File). Based on these few sources it remains unclear whether Bactrian camels would benefit from a less seasonal breeding to minimize the potential time lost between breeding seasons. Arguably, long gestation periods and a certain seasonality that prevent an 'optimal' reproductive output, together with a relatively low metabolism and a limited food intake capacity in camelids [111,112], may contribute to comparatively long generation times that make camelids non-competitive in resource-rich environments [113].

In SAC, lactation lasts about six months for all species (S2 Table in S1 File). A specialty was seen in guanacos, where both newborns and yearlings were nursed [33,38], although a first weaning was observed 7 months postpartum [33]. For all SAC, lactation would thus fit within an annual reproduction.

In seasonal environments, energetically expensive activities such as lactation are typically restricted to seasons with favourable environmental conditions – usually late spring/ early summer [114]. Therefore, the tendency to short-day breeding in SAC is astonishing, as it shifts births into the mid to late summer. However, this short-day breeding tendency might be explained by the habitat characteristics of SAC.

Vicunas inhabit the "Puna" – a high-altitude grass- and scrubland in the Andes (3700–4900 m a.s.l., 9°-29° S) [38,115]. The Puna is a cold, arid region with ¾ of the annual precipitation falling in austral summer (December to February) [38,53,115–117]. Although austral summer is mildly warmer than the rest of the year and cloud cover often prevents freezing night temperatures [38], precipitation still mostly falls as hail and snow. Nevertheless, the vegetation grows and becomes green and water is widespread after austral summer [53]. After the increased precipitation in austral summer, soil and vegetation become dry again towards austral winter. Austral Spring brings warmth to a ground already dry and there is hardly any vegetational growth – unlike most other regions of the world [38,53].

Similar to vicunas, guanacos are native to arid environments in the Andes and Patagonia (from sea level up to over 4000 meters, 8° to 55° S) [37,38,115]. Due to this broad latitudinal distribution, it is impossible to summarize the climate of each habitat. Most studies were held either on the island of Tierra del Fuego in the southeastern end of the South American continent or in the Torres del Paine National Park in southern Chile. On the Island of Tierra del Fuego, precipitation falls year-round [69], but is a bit more concentrated during austral autumn and early winter [37]. The austral winters get extremely dry if there is no snow cover [38] and the strongest winds are present in austral spring and summer [37]. The growing season is during austral summer (December to February) [38]. In the Torres del Paine National Park, 60% of the precipitation falls between January and May (austral summer/ autumn), with a warmer, windy and rainy season from October to April and a cold, relatively dry season from May to September [118].

These habitats of guanaco and vicuna share a cold and dry austral winter and vegetation often does not grow before austral summer; additionally, the strongest winds are present in austral spring and summer. In spring, when most mammals in other parts of the world have their young, the habitats probably do not yet provide enough resources and are dried out. Further, SAC do not lick their newborns dry [10], therefore the young are particularly prone to hypothermia in harsh climatic environments. This might explain why guanacos and vicunas show a tendency to short-day breeding. In vicunas, conceptions peaked just after the rainy season which could indicate a body-condition driven seasonality. However, the fact that this peculiar seasonal pattern persists in captivity well out of the natural habitat – where spring would represent the evident birthing period of choice, and where resources generally are unlimited – suggests a photoperiod triggered seasonality and not one mainly dependent on body condition [3]. In dromedary camels, local rainfall and conception overlap at least partially, but in a few regions the rainy season falls in between two consecutive breeding seasons (sources h to l in Table 6). For guanacos, llamas and Bactrian camels a possible correlation between rainfall and conception is not visible (Tables 3–5).

We provide a detailed review on breeding seasonality in camelids based on both literature and conceptions in zoos. Zoo populations could only be reviewed as a whole and not in individual populations per origin. As we compared

conceptions, we often had to calculate back from births, leading to minor errors due to standardized gestation lengths of whole months. In addition, conceptions (or births) dates were typically only recorded by month, making a conception on the 1st or the 31st of a month identical in our data. Interpretations of conceptions in relation to, e.g., the summer solstice were therefore less precisely feasible. The unexpected finding of a tendency to a short-day breeding season in SAC should be further investigated in future studies. Possible underlying triggers like diet or the local climatic conditions in the habitats should be a main focus.

## Conclusions

In OWC (*C. bactrinaus* and *C. dromedarius*) a tendency towards long-day breeding was found, which was more distinct in Bactrian camels. Similar conception peaks in literature and zoo populations strongly suggest an underlying photoperiodic trigger. Nevertheless, both species were also able to breed year-round, which was more pronounced in zoo populations. OWC share a gestation length of around 13 months and were expected to be less seasonal than SAC because they would lose a whole reproductive year every other year under strict seasonality. However, Bactrian camels were more seasonal than dromedaries (and vicunas and guanacos were more seasonal than alpacas and llamas). Lactation could act as a "filler" between two consecutive births and would mitigate the negative effect of a strictly seasonal breeding in OWC. Duration of lactation in literature was too various (7–24 months) to draw firm conclusions.

SAC (*L. glama, L. guanicoe, V. pacos* and *V. vicugna*) unexpectedly showed tendencies to a short-day breeding season, both in literature and zoo populations. Like OWC, they nevertheless were also able to breed year-round. An explanation for this breeding pattern might be in the climatic conditions of their natural habitats. Vicunas and guanacos live in dry grasslands with main rainfall in austral summer, which initiates vegetational growth. A breeding seasonality influenced by a body-condition threshold is a possibility based on these vegetational growth patterns; however, we observed the same conception peaks both in wild and zoo populations, which again strongly suggests an underlying photoperiodic trigger. Domesticated SAC showed a weaker seasonality than nondomestic taxa. To what degree SAC might be limited in their geographic range by their peculiar reproductive seasonality that is tuned to characteristics of their specific habitats is an open question.

The mixed breeding pattern in OWC and SAC, where distinct seasonal conception peaks and the capacity for breeding all year round exist together, could derive from two different scenarios. Either the phototrigger is active in all specimens, but at a level that can be easily overruled. Or populations consist of different phenotypes, as previously mainly described in rodents, with some photo-triggered individuals and some that are photo-independent. Detailed studies are required to decide between these hypotheses.

## Supporting information

**S1 Data. Pauli_CamelidConceptionSeasonality_ZooData.**
(XLSX)

**S1 File. SeasonalityCamelids_SupplementsClean_251121_mc.**
(DOCX)

## Acknowledgments

This research was made possible by the worldwide information network of zoos and aquariums which are members of Species360 and is authorized by Species360 Research Data Agreement #2019-Q3-RR4. We thank Barbara Schneider for tireless support in literature acquisition, and Jeanne Peter for drawing the animal icons in Fig 1. MC developed the concept and supervised the work, MR analysed the zoo data, NP performed the literature research, literature data evaluation, and wrote the first draft of the manuscript that received input from the other authors.

## Author contributions

**Conceptualization:** Marcus Clauss.

**Data curation:** Nicole Pauli, Marco Roller.

**Formal analysis:** Nicole Pauli.

**Investigation:** Nicole Pauli.

**Methodology:** Marcus Clauss.

**Project administration:** Nicole Pauli.

**Resources:** Marco Roller.

**Supervision:** Marcus Clauss.

**Validation:** Marcus Clauss.

**Visualization:** Nicole Pauli.

**Writing – original draft:** Nicole Pauli.

**Writing – review & editing:** Marco Roller, Marcus Clauss.

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
