## [Decision Letter · Decision Letter 0]

6 Nov 2025

Dear Dr. Clauss,

Thank you for submitting your manuscript to PLOS ONE. After careful consideration, we feel that it has merit but does not fully meet PLOS ONE’s publication criteria as it currently stands. Therefore, we invite you to submit a revised version of the manuscript that addresses the points raised during the review process.

We look forward to receiving your revised manuscript.

Kind regards,

Gisele Akemi Oda, Ph.D.

Academic Editor

PLOS ONE

Journal Requirements:

2. We note that Figure 1 in your submission may contain copyrighted images. All PLOS content is published under the Creative Commons Attribution License (CC BY 4.0), which means that the manuscript, images, and Supporting Information files will be freely available online, and any third party is permitted to access, download, copy, distribute, and use these materials in any way, even commercially, with proper attribution. For more information, see our copyright guidelines: http://journals.plos.org/plosone/s/licenses-and-copyright.

Please upload the completed Content Permission Form or other proof of granted permissions as an 'Other' file with your submission.

3. Please upload a copy of Figure 1, to which you refer in your text on page 14. If the figure is no longer to be included as part of the submission please remove all reference to it within the text.

4. Please include a caption for figure 6.

Reviewers' comments:

Reviewer's Responses to Questions

**Comments to the Author**

1. Is the manuscript technically sound, and do the data support the conclusions?

Reviewer #1: Yes

2. Has the statistical analysis been performed appropriately and rigorously?

Reviewer #1: N/A

3. Have the authors made all data underlying the findings in their manuscript fully available?

Reviewer #1: Yes

4. Is the manuscript presented in an intelligible fashion and written in standard English?

Reviewer #1: Yes

Reviewer #1: This is a thorough review of breeding seasonality in camelids. Data collection appears exhaustive and the evaluation and interpretation of the published data careful. However, I found it hard to follow the authors’ text, because they do not define the seasons in the Method section and in the tables always refer to increasing or decreasing daylength, not seasons. In the text to the various species we are then told that breeding occurs mainly in autumn or spring. From the Tables I take it that autumn refers mainly to March, April and May in the South and Aug, Sep, and Oct in the North (??). This needs to be made explicit for all seasons in the Method section. Also, clearly define what are long or short days and how this relates to increasing or decreasing daylength.

Also, I consider the reasoning (for example line 82) that species with a MIBL >12 months can be expected to be less seasonal breeders a bit too one-dimensional. If the ecology is sufficiently predictably bad or excellent (for mother or offspring) at certain seasons they will be forced into seasonal breeding no matter what the MIBL.

Some discussion of the phylogenetic relationship of these species might also help. It is to be expected that Vicuña and Alpaca and Guanaco and Llama, respectively, are more similar to each other in their response to photoperiod simply because of their ancestry. Are there any data on C. ferus (or C. bactrianus ferus as others classify it) from China or Mongolia?

I found Fig. 1 not particularly helpful: The day length cycle between 10 and 17 hours is specific to a certain latitude and the selection of species shown appears rather arbitrary.

Minor comments

L 35 here the ambiguity about definitions of day length surfaces: If both groups are expected to be long day breeders (which I interpret to be perhaps May to August in the Northern and November to February in the Southern hemisphere, then offspring would be borne either into long or decreasing daylengths again for SAC) or into autumn for OWC. Why should this lead to births in spring?

L 27, 53 adaption? Why not adaptation?

L 110 not necessarily long day but perhaps better “increasing daylength” breeders

L 181 correct scientific name is Vicugna vicugna (correct also in Fig. 2)

**Do you want your identity to be public for this peer review?** For information about this choice, including consent withdrawal, please see our Privacy Policy

Reviewer #1: No

---

## [Author Response · Author response to Decision Letter 1]

26 Nov 2025

please see the attahed Response to Reviewers document (a table I cannot paste here in the online system)

---

## [Editor Report · Decision Letter 1]

7 Dec 2025

Breeding seasonality of Tylopoda: expected patterns in Old World camelids but an exceptional pattern in South American camelids

PONE-D-25-51657R1

Dear Dr. Clauss,

We’re pleased to inform you that your manuscript has been judged scientifically suitable for publication and will be formally accepted for publication once it meets all outstanding technical requirements.

Kind regards,

Gisele Akemi Oda, Ph.D.

Academic Editor

PLOS One

---

## [Editor Report · Acceptance letter]

PONE-D-25-51657R1

PLOS One

Dear Dr. Clauss,

I'm pleased to inform you that your manuscript has been deemed suitable for publication in PLOS One. Congratulations! Your manuscript is now being handed over to our production team.

Kind regards,

on behalf of

Dr. Gisele Akemi Oda

Academic Editor

PLOS One